# Pangenomics of the Symbiotic Rhizobiales. Core and Accessory Functions Across a Group Endowed with High Levels of Genomic Plasticity

**DOI:** 10.3390/microorganisms9020407

**Published:** 2021-02-16

**Authors:** Riccardo Rosselli, Nicola La Porta, Rosella Muresu, Piergiorgio Stevanato, Giuseppe Concheri, Andrea Squartini

**Affiliations:** 1Department of Marine Microbiology and Biogeochemistry, NIOZ Royal Netherlands Institute of Sea Research, NL-1790 AB Den Burg, The Netherlands; riccardo.rosselli@gmail.com; 2Departamento de Fisiología, Genética y Microbiología, Universidad de Alicante, 03690 Alicante, Spain; 3Department of Sustainable Agrobiosystems and Bioresources, Research and Innovation Centre, Fondazione Edmund Mach, 38098 San Michele all’Adige, Italy; nicola.laporta@fmach.it; 4MOUNTFOR Project Centre, European Forest Institute, 38098 San Michele all’Adige, Italy; 5Institute of Animal Production Systems in Mediterranean Environments-National Research Council, 07040 Sassari, Italy; ro.muresu@tiscali.it; 6Department of Agronomy, Food, Natural Resources, Animals and Environment, University of Padova, 35020 Legnaro, Italy; stevanato@unipd.it (P.S.); giuseppe.concheri@unipd.it (G.C.)

**Keywords:** pangenome, core genome, rhizobiales, nitrogen-fixing symbiosis, nodulation genes, functional divergence, *Rhizobium sullae*

## Abstract

Pangenome analyses reveal major clues on evolutionary instances and critical genome core conservation. The order Rhizobiales encompasses several families with rather disparate ecological attitudes. Among them, Rhizobiaceae, Bradyrhizobiaceae, Phyllobacteriacreae and Xanthobacteriaceae, include members proficient in mutualistic symbioses with plants based on the bacterial conversion of N_2_ into ammonia (nitrogen-fixation). The pangenome of 12 nitrogen-fixing plant symbionts of the Rhizobiales was analyzed yielding total 37,364 loci, with a core genome constituting 700 genes. The percentage of core genes averaged 10.2% over single genomes, and between 5% to 7% were found to be plasmid-associated. The comparison between a representative reference genome and the core genome subset, showed the core genome highly enriched in genes for macromolecule metabolism, ribosomal constituents and overall translation machinery, while membrane/periplasm-associated genes, and transport domains resulted under-represented. The analysis of protein functions revealed that between 1.7% and 4.9% of core proteins could putatively have different functions.

## 1. Introduction

With increasing availabilities of sequenced genomes in databases, pangenome-wide analyses have become a valuable tool to explore the breadth of genetic variability and capture the critical core-genes subset within a clade [1,2]. The boundaries of this comparison could be set at species level, in which a number of strains are analyzed, as in the cases of *Escherichia coli* [3], *Sinorhizobium meliloti* [4], *Pseudomonas aeruginosa* [5], or at higher rank level, as for the genus *Clostridium* [6] or *Rhizobium* [7]. Furthermore, comparisons can cross the hierarchical taxonomy and focus on common physiological/ecological features of distantly related organisms, as shown in the study of the predatory bacteria pangenome [8]. This functional group includes as many as 11 different lineages that occur throughout diverse unrelated habitats and has revealed interesting clues on the importance of gene acquisition processes in the adaptation of those species. In a recent study, the horizontal DNA transfer among 27 rhizobial genomes was analyzed and the resulting patterns concurred with the view that rhizobia do adapt to the host environment through a balance of loss and gain of symbiosis genes through strong purifying selection [9].

In fact, prokaryotes are largely subjected to horizontal gene transfer that spans systematics ranks and determines common phenotypic traits, another example of which is the minimum gene set required for either for access or interaction between bacteria and higher plants which is also conserved between distant related organisms [10]. Another objective of comparative genomics between distant related taxa is to analyze shared genes under the light of possible functional divergences. The proof of concept follows the premises of Kimura’s neutral theory of evolution. According to this postulate, the rate synonymous vs. non-synonymous nucleotide substitutions and subsequent amino acidic variations can be indicative of genes undergoing selective pressure. Therefore, a cutoff value can be set to predict an achieved bona fide functional divergence [11]. Extended comparative studies that included bacterial genomes/proteomes, and updated computational tools to score the putative functional divergence have been proposed and applied [12].

Fitting the aspect of protein functional divergence within a pangenome study could appear contradictory. If the core genome consists, by definition, of orthologous genes (which in principle are likely to conserve the same function), logically there could not be a functional divergence. This apparent paradox has been overcome by different authors who concluded that many orthologs could actually bear different functions [13,14,15,16].

As a dataset we selected a group of symbiotic nitrogen-fixing bacteria of primary importance for overall plant ecology as well as agricultural productivity. The bacterial order of Rhizobiales (Alpha class of the Proteobacteria phylum) is a taxonomically wide group that currently harbors 15 families, within which, few species possess the capability of engaging nitrogen-fixing symbiotic relationships with legume plants. The order is overtly polyphyletic. Analyses of small ribosomal subunit sequences [17] showed the presence of several non-plant associated genera (*Rickettsia*, *Rochalimaea*, *Brucella*, *Erythrobacter*, *Rhodospirillum*, *Rhodopseudomonas*) intermingled within the 16S dendrogram’s terminal branches with *Rhizobium*, *Bradyrhizobium*, *Mesorhizobium*, *Sinorhizobium*, *Azorhizobium*, and alike. This configuration made clear that the Rhizobiales evolutionary line could not be traced as a vertical, chronologically-coherent, cell division-related path, but rather as a network of lateral gene-transfer processes, mediated primarily by large mobile plasmids and fostered by abundant insertion elements [18]. The nitrogen-fixing symbiotic members of Rhizobiales, in spite of this definition based on the specific metabolic pathway, display a variety of differences in terms of growth kinetics (from fast- to slow-growing), basic physiology (from phototrophy to heterotrophy), genome allocation (from all-chromosomal, to split in up to six plasmids), host plant specificity (from strictly monospecific to highly promiscuous) and nodulation site (from root- to stem-nodules) [19,20,21]. In addition, some rhizobia as *R. sullae* [22] have hosts that can still be found both in wild and in cropped status [23,24]. All such premises make it particularly intriguing to explore the features of this group’s pangenome, to examine the boundaries and extent of the core genome, and finally the possible instances of functional divergence across orthologous genes.

The goal of this work was, therefore, to trace a picture of the balance between gene conservation and gene evolution within a group of bacteria characterized by very dynamic genetics as well as by a broad eco-physiological latitude span.

In particular, the rationale was to test the degree of genomic divergence in spite of the common, phenotypically unifying, trait of legume nodulation.

Our hypotheses were that (a) the genomic plasticity of Rhizobiales would suit the need for tracing an efficient resolution of their differences based on nucleotide and amino acidic average identities across genomic data and (b) that different functional protein categories would clearly partition as over- or under-represented between the core and the flexible subsets of their genomes, reflecting host-specific adaptations.

## 2. Materials and Methods

### 2.1. Selected Bacterial Strains and Main Genomic Features

The dataset of species to be analyzed was chosen by using the following criteria: (i) available genomes that were selected all belong to the Rhizobiales order, and (ii) these organisms are nitrogen-fixing symbionts of plants. Under these constraints, among the existing possibilities, we selected 12 strains belonging to different rhizobia clades to be included in the present analysis. Their features are described hereafter.

*Azorhizobium caulinodans* ORS571; Family: Xanthobacteraceae; GenBank accession: GCA_000010525.1; genome size: 5,369,772 nucleotides [25]; number of plasmids: 0. Fast growing, isolated from *Sesbania rostrata* in the Sahel region of Africa [26]. *A. caulonidans* has the peculiarity of nodulating also on the stem, as the symbiont plant grows in habitats that are often submerged. Its host range is restricted to the host of isolation.

*Bradyrhizobium japonicum* USDA 110 (also known as *B. diazoefficiens* USDA 110); Family: Bradyrhizobiaceae; RefSeq Accession: GCF_000011365.1; genome size: 9,105,828 nucleotides [27]; number of plasmids: 0. Slow growing, isolated from the soybean *Glycine max* in Florida, USA. This strain of *B. japonicus* nodulates on plant roots. Nodulation is host-specific and moreover limited to defined serogroups.

*Bradyrhizobium japonicum* USDA 6^T^; Family: Bradyrhizobiaceae; GenBank accession: GCA_000284375.1; genome size: 9,207,384 nucleotides [28]; number of plasmids: 0. Slow growing, isolated from soybean in Japan. The strain of *B. japonicum* USDA 6^T^ is a host-specific symbiont that shares extended portions of genomic collinearity with *B. japonicum*, USDA 110 but a large region of genomic inversion. The nodulation occurs at the level of the root system. It is the type strain of the species.

*Bradyrhizobium* sp. BTAi1; Family: Bradyrhizobiaceae; GenBank accession: GCA_000015165.1; genome size: 8,493,513 nucleotides [29]; number of plasmids: 1. Slow growing, isolated from stem nodules of the aquatic legume *Aeschinomene indica* in North America. *Bradyrhizobium* sp. BTAi1 is capable of performing anoxygenic photosynthesis, denitrification, and it lacks the common nodulation genes which are shared by the vast majority of known rhizobia. The nodulation can occur at either roots or stems of specific plant hosts.

*Bradyrhizobium* sp. ORS278; Family: Bradyrhizobiaceae; GenBank accession: GCA_000026145.1; genome size: 7,456,587 nucleotides [29]; number of plasmids: 0. Slow-growing, isolated from stem nodules of *Aeschinomene sensitiva* in Senegal, Africa. It shares the peculiarities described for strain BTAi1 (anoxygenic photosynthesis, denitrification, and lack of nod genes) to which is highly similar but lacks a large plasmid. Nodulation occurs on roots or stems of specific plant hosts.

*Mesorhizobium loti* MAFF303099 (also known as *Mesorhizobium huakuii* bv. *loti*) Family: Phyllobacteriaceae; Genbank Accession: GCA_000009625.1; genome size: 7,596,297 nucleotides [30]; number of plasmids: 0. Fast growing, isolated from *Lotus japonicus* in Japan. Nodulation on roots. The host range is wider than that described for the prior cases and includes the genera *Lotus*, *Anthyllis* and *Lupinus.*

*Rhizobium leguminosarum* bv. *viciae* 3841; Family: Rhizobiaceae; Genbank Accession: GCA_000009265.1; genome size: 7,751,309 nucleotides [31]; number of plasmids: 6. Fast growing; isolated from peas (*Pisum sativum*) in England. Nodulation on roots. The host range includes plants belonging to the genera *Pisum*, *Vicia*, and *Lens.*

*Rhizobium etli* CFN42^T^ Family: Rhizobiaceae; Genbank Accession: GCA_000092045.1; genome size: 6,530,228 nucleotides [32]; number of plasmids: 6. Fast growing; isolated from bean plants of the species *Phaseolus vulgaris* in Mexico. Nodulation on roots. The host range includes species within the single genus *Phaseolus.* It is the type strain of the species.

*Rhizobium sullae* IS123^T^ Family: Rhizobiaceae; Genbank Accession: GCA_900169785.1; genome size: 7,889,576 nucleotides [33]; number of plasmids: not specified. Fast-growing, isolated from wild stands of *Sulla coronaria* (formerly *Hedysarum coronarium*) in Spain. The host plant, widely diffused in the Mediterranean countries, has the peculiarity of existing still as native weed in wild conditions as well as agriculturally cropped ecotypes. Nodulation on roots; Host specific to *H.coronarium.* It is the type strain of the species.

*Rhizobium sullae* WSM1592 Family: Rhizobiaceae; Genbank Accession: GCA000427985.1; genome size: 7,530,820 nucleotides [34]; number of plasmids: not specified. Fast growing, isolated from a cropped variety of *Hedysarum coronarium* on Sardinia island (Italy). Nodulation on roots of the specific symbiont plant *H. coronarium.*

*Sinorhizobium fredii* NGR234; Family: Rhizobiaceae; RefSeq Accession: GCF_000018545.1; genome size: 6,891,900 nucleotides [35]; number of plasmids: 2. Fast growing, isolated from *Lablab purpureus* in New Guinea. Nodulation on roots, this strain is able to establish a nitrogen fixing symbiosis with more than 112 plant genera including the non-legume *Parasponia* [36]. Currently no other rhizobial strain is known to exhibit such extremely wide host promiscuity.

*Sinorhizobium meliloti* 1021; Family: Rhizobiaceae; RefSeq Accession: GCF_000006965.1; genome size: 6,691,694 nucleotides [37]; number of plasmids: 2. Slow growing, isolated from *Medicago sativa* in France. Nodulation on roots of multiple hosts whit a range including the genera *Medicago, Melilotus, Trigonella.*

### 2.2. Bioinformatics Analyses

Average nucleotide identity (ANI) and average amino acid identity (AAI) between Rhizobiales proteomes (i.e., predicted genes and proteins) were computed as previously described [38,39]. The pan-genome analysis of the references was performed using the software GET_HOMOLOGUES [40], implemented with the -M option which applies the orthoMCL algorithm to infer reciprocal homologies between genes. According to the results provided by such software, genes belonging to the strict-core [41] were separated by the others and analyzed in further depth. A phylogenomic tree was produced for the core gene set of the analyzed genomes using the PhyloPhlAn software tool, version 3.0 [42], with the Phylophlan database for broad bacteria comparisons.

A whole-genome orthologs search was performed to identify conserved functions across the different organisms using the Reciprocal Best BLAST Hit (RBBH, version 0.1.11) [43] setting an E-value cutoff of 1 × 10^−5^. Genes with a minimum of 60% identity and coverage higher than 60% were considered as orthologous candidates.

In addition to the aforementioned analyses, the core genome of Rhizobiales was analysed using a HMM (Hidden Markov Model) similarity search approach implemented in the bcgTree pipeline [44]. Genes which were confirmed as core genome components were aligned back to all genomes under study in order to score for conservation percentage.

The MAPLE resource (Metabolic and Physiological Potential Evaluator) [45] was used to estimate function abundance and evaluate metabolic and physiological potential [46]. The reference database was KEGG (Kyoto Encyclopedia of Genes and Genomes) [47], and proteins were mapped and normalized on the ribosomal proteins counts in its pathway database. KAAS (KEGG Automatic Annotation Server) was used for ortholog assignment (KO, KEGG Orthology) and pathway mapping [48] Functional divergence was assessed using DIVERGE 2.0 software [49]. Output data were parsed by ad hoc Perl scripts, while R provided statistical and graphic support.

## 3. Results

### 3.1. Phylogenetic Relatedness Among the 12 Rhizobiales Strains

To frame the systematics position of the 12 pure isolates compared, ANI and AAI were computed using the whole genome information (Figure 1). The span of difference resulted from a minimum of 78% to a maximum of 99% ANI, which corresponded to minimum 50% to the maximum of 95% AAI.

Top ANI and AAI values were found between *Rhizobium* species, as well as between species belonging to the genus *Bradyrhizobium*. Highest ANI and AAI were found between *R. sullae* IS123^T^ and *R. sullae* WSM1592 (~99%, ~95%), pointing that these two strains are the most closely related pair of the analysis. Within *Bradyrhizobium* species, it should be noted that ~92% ANI and ~87% AAI between the two *B. japonicum* species indicated that genetic and potential functional drift occurred in spite of extended and conserved collinearities at the level of their genomes. Interestingly, both *Rhizobium sullae* and *Bradyrhizobium japonicum* strain pairs were isolated from different places (Italy and Spain for *R. sullae*; USA and Japan for *B. japonicum*), and both genera show high specificity for their host macro-symbiont. This might imply that selective pressures differentially reshaped their genomes, favoring the divergence in case of *Bradhyrhizobium japonicum* and promoting conservation in the case of *Rhizobium sullae*. This difference can be explained by the fact that the soybean host of *B. japonicum* was originally confined in far eastern Asia and its presence elsewhere, including America, is the result of man-managed cropping. On the contrary, both Italy and Spain are within the native home range of *Sulla coronaria* which is an Eurimediterranean plant.

### 3.2. Pangenome and Core Genome Analyses

A total number of 84,482 genes was included in the analysis and comprised genes encoded either by the chromosome or by plasmids in the genomes of Rhizobiales. After removing duplicates, i.e., paralog genes in the genome, the reference pangenome was reduced to 37,364 non-redundant loci.

Concerning the core genome across the 12 taxa which was retrieved by GET_HOMOLOGUES, it was found to be constituted by 700 genes, which corresponded to 1.87% of the pangenome (the full list of genes that belong to the core genome is available in Appendix A and the comparisons between core genome and whole genome are provided in Appendix A). The accessory (flexible) genome, defined by genes which were not unique but not shared by all strains considered, and thus not included in the core genome [50], was equal to 15,394 genes. The number of unique genes (i.e., those that were identified as strain-specific) was in total 21,970 and accounted the 58.8% of the pangenome. The percentage of core genes over the number of total genes in each of the single genomes ranged from a minimum of 7.88% (*B. japonicum* UDSDA 6) to a maximum of 15.64% (*A. caulinodans* ORS571), with an average value per genome of 10.2%.

The gene partitioning on chromosomal or extra-chromosomal replicons in each of the genomes and its relations with the core genome showed differences at the level of different Rhizobiales species and strains. The details are presented in Table 1.

To verify whether genes belonging to the core genome were still supportive of the taxonomic distances resulting from the ANI of full Rhizobiales genomes, a phylogenomic tree was constructed using solely the 700 genes which were identified as core genome components (Figure 2). The topology obtained was consistent with the overall systematic placement of each strain at high taxonomy level, nevertheless different relationships were found regarding the position of *A. caulinodans* ORS57 with respect to the other Rhizobiales. The comparison between whole genomes, indeed, placed this microorganism close to the clade constituted by *M. loti* MAFF303099 and the two *Sinorhizobium* species. Considering the gene subset belonging to the core genome (Figure 2), *A. caulinodans* ORS57 appears closer to the clade which includes the four *Bradyrhizobium* species here considered. We speculate that core and flexible genomes followed two distinct paths. The core genome appears more conserved in specific Rhizobiales groups, such as *Azorhizobium*, and *Bradhyrizobium.* The flexible genome would have been more likely reshaped by evolutionary shortcuts due to horizontal gene transfer, which would have brought higher similarities among *Azorhizobium, Mesorhizobium* and *Sinorhizobium*. An unrooted phylogenomic topology showing relationships between Rhizobiales basing on core-gene comparisons is instead shown in Appendix A.

The analysis conducted through RBBH approach showed that *Azorhizobium caulinodans* ORS571 and the two *Sinorhizobium* species, *S. fredii* and *S. meliloti,* shared the lowest number of orthologous genes. As expected, the most correlated species belonged to same bacterial lineages and were identified to be the different *Bradyrhizobium japonicum* species and *Bradyrhizobium* sp. BtAi1 and ORS278. The Pearson correlation coefficient based on the RBBH result, showed correlation values comprised between -0.94 and 1 (Figure 3, panel A), and was in general consistent with the clustering of Rhizobiales genomes based on the ANI. Coefficient values visually separated Rhizobiales into two main clades, one which included *Rhizobium* and *Sinorhizobium* species (correlation coefficients comprised between 0.86 and 1), and the other which included all *Bradyrhizobium* species and *Azorhizobium caulinodans* ORS571 (coefficients between 0.85 and 0.95). In addition, the analysis of each genome percentage, which corresponded to conserved orthologous in the specific set of pairwise genomes, comprised between 49–75% between *Bradyrhizobium* species (average 58.3%), and it was just slightly lower (average 54.6%) between *Rhizobium* and *Sinorhizobium*. The relative genome percentage, which resulted constituted by proteins encoded by ortholog genes, was therefore comparable between distinct Rhizobiales groups, although the latter were constituted by organisms belonging to different species. The numbers of shared genes for each pairwise comparison is shown in the Appendix A.

Regarding the degree of in-genome duplication, which corresponds to the generation of paralog genes, the first selection of the core genome included the filtering-off of those genes which belonged to a same gene family. Nevertheless, we sought to envisage the extent of this phenomenon and analyze the corresponding number of paralog genes within each genome (Table 2). The most frequent cases of paralogs were originated by single-step gene duplication (2 copies of the gene per genome), but in some species (mostly Bradyrhizobiaceae) cases of a multiple-duplication of a given gene with up to 13 copies were observed.

### 3.3. Analysis of Core Genome Function Enrichment and Depletion Using a Single Reference Genome

Having established the size of the 12 symbiotic Rhizobiales core genome, in order to ascertain whether such critically conserved gene subset would be related to particular biological processes, we used a reference-driven conserved genes estimation strategy. In essence, we correlated a full representative genome with predicted functions identified among the core genome genes. The functional analysis was performed by compiling the corresponding tables for the gene categories and Gene Ontology (GO) terms, distinguishing the cellular component, biological process and molecular function items, drawn from the corresponding genome project annotation. As reference genome for this analysis, we chose *Rhizobium leguminosarum* bv. *viciae* 3841 and plotted the data for the full genome (7342 genes) and for the core genome containing 700 of its genes (Table 3). Data showed that the top represented categories encoded by the *R.leguminosarum* 3841 genome include genes putatively involved in metabolism and processing of small molecules (~21% of the total genes) and related to cellular processes (~20%). Nevertheless, although the first of these categories also constituted the majority of shared genes with other Rhizobiales, the second most abundant function is represented by metabolism of macromolecules (39% of the core genome).

Looking at the detailed GO terms, we analyzed the difference between the percentage of proteins for each GO annotation in the reference genomes and the percentage of those proteins at the level of the core, considering as significant those categories which were up- or down-represented above of below 1%. The comparison showed that the top enriched categories (i.e., positive percentage difference at the level of the core genome) regarding Cellular Component GO included overall proteins involved in the cytoplasm metabolism (Figure 4). They constituted 9.6% of the reference *R. leguminosarum* bv. *viciae* 3841 genome and represent 21% of the core genome (+12%). The second most represented category was found to be related to ribosome metabolism (2.5% of the reference genomes, 9.75% of the core), suggesting a conservation of the transcription-related metabolism among Rhizobiales bacteria. Among the less-represented functions related to this GO category, membrane-related components appeared strongly depleted in the core genome. Although 729 proteins belonged to this GO group, resulting in the most represented category in the reference genome (36% of total genes with GO term), they constituted 14% of the core proteins (-22%). Consistent with the low representation of membrane related proteins in the core functions of Rhizobiales, also proteins integral to membrane and proteins bound to the periplasmic space were less represented in the core genome (7.51% and 1.2%, respectively), in spite of the fact that both were among the top 10 most represented categories in the reference genome.

Concerning proteins which function was related to broad biological processes, enriched categories in the core genome were frequently associated to basal metabolism and housekeeping functions related to DNA-activity and genome maintenance (Figure 5). The top enriched category in the core genome was related to translation machinery, which constituted the 1.6% of total proteins in the reference genome and corresponded to the 7.25% of the core (+5.58%), followed by DNA-repair (+1.68% in the core genome) and rRNA-metabolism (1.17%). Consistent with high variability of membrane-associated components between Rhizobiales bacteria, among the top depleted categories in the core genome resulted different transport-related proteins (transport components, –8.84% occurrences in the core genome; transmembrane transport, –1.82%) and proteins involved in regulation of the gene expression transcription regulation, –1.95%).

Similar results were found also for the category related to the molecular function (Figure 6), in which ribosomal components, RNA-and ATP-binding proteins were conserved core-genome constituents (between +1.38% and +2.17%). Proteins which have transport activity, and that are likely to be associated with cellular membranes, were those which showed higher species-specificity and were less represented in the core genome (−5.45%). The low representation of proteins whose function is related to transcription processes, including transcription factors and DNA-binding proteins, suggests adaptation at different lifestyles, probably related to different host specificities and types of symbiotic relationship.

### 3.4. Functional Divergence Analysis

Besides the above described pangenome analysis, we carried out a pairwise comparison of each of the 12 isolates (in total, 144 matches), checking the possible extent of functional divergence between each shared orthologous gene belonging to the core genome. The objective of this analysis was to assess the magnitude of the genetic drift that could have determined a functional variation.

Between 19 and 208 instances scoring significantly as bona fide functionally divergent genes were found (Appendix A). These genes corresponded to the minimum 1.39% (*A. caulinodans* vs. *R. sullae* WSM 1592) to the maximum 4.48% (*R.etli* vs. *R. leguminosarum*) of shared orthologous genes between Rhizobiales genomes. The top list (the eight highest-scoring cases) of these genes, displaying a high-frequency of parallel divergence across several taxa, is shown in Table 4, along with the corresponding gene function annotations. Analyzing in depth all the diverged genes, we noticed that some among them encoded housekeeping proteins which were not expected to being subjected to functional divergence. For example, the gene encoding the DNA polymerase I was found to be a divergent candidate in the four *Bradyrhizobium* species well as *A. caulinodans* ORS571. Another gene which the analysis scored as putatively divergent encodes the gamma chain of the ATP-synthase, and data indicated that such as functional variation would have occurred in three out of the four *Bradyrhizobium* genomes. Other genes were involved in different metabolic pathways (e.g., cobyrinic acid synthase) and metabolisms (e.g., glutathione synthetase). Notably, these genes corresponded to same orthologous in more than two rhizobia genomes. It could be argued that these genes could have independently acquired functional divergence in several different rhizobia.

## 4. Discussion

The pangenome of the taxonomically heterogeneous group of Rhizobiales investigated here displayed a particularly large size, amounting to 37,364 gene families, and a nearly 50-fold smaller core genome (1.87 % of the pangenome). To give some comparisons, an intra-species pangenome with three strains which was reported for *S.meliloti* had a core genome of 5075 orthologous groups while those groups which constituted the flexible genome were 3810 [51]. The analysis of predatory bacteria [8], which was performed by testing 11 species, classified only 1977 gene families with a core genome constituted of 509 units. Our data, therefore, strengthen the evidence that symbiotic rhizobial diversity is represented by transversal events of gene acquisition moving genetic islands across species of rather distant ancestry [7,18].

The pairwise shared genes calculation yielded results that can be compared to the standard genome-to-genome alignments as ANI and AAI. In relation to the latter two, it can be commented that the pairwise orthologous presence analysis offers a resolution higher than each of those contributed by ANI or AAI, going from a minimum of 15.57 % (*R. etli* vs. *B. japonicum* USDA6) of conserved genes over the complete genome, to a maximum of 86.33 % (*R. sullae* IS123^T^ vs. *R. sullae* WSM1592). The 71% difference between these two extremes defines the interval which could be considered the index of resolution. The corresponding minimum and maximum ANI values were 68% and 98%, with an interval at 30%, while for AAI values (minimum 49%, maximum 82%) the fork spanned across the 33% difference.

The analysis of the number of paralogs showed a degree of variability that amounted from a minimum of 0.75% of the genes present in the full genome of *A. caulinodans* ORS571to the maximum 3.22% recorded in the genome of *S. fredii* NGR234. As expected, an analysis of the annotated functions of the paralogous genes revealed that all those genes that had high degrees of duplication (above 7 copies per genome) corresponded to transposase-encoding genes. This finding highlights the fact that many Rhizobiales genomes contain a particularly abundant set of insertion elements, which are regarded as a major source of genomic evolutionary plasticity [52,53,54]. Further supporting the magnitude of extrachromosomal inheritance in shaping the genome architecture of this clade, it is worth noticing that 5–7% of the core genes map on plasmids. Indeed, this testifies the relevance of mobile replicons in spreading core genes among N-fixing symbionts. Arguably, the whole phenotypic complex of plant symbiotic N-fixing capability could have been moved by plasmids. This instance was advanced early by authors that analyzed the Alphaproteobacteria 16S-based taxonomy trees [17]. These analyses reckoned that the last possible common ancestor of plant symbiotic N-fixing rhizobia would have existed in an era (> 500 MY before present) in which plants were just barely conquering land. As a consequence, the onset of root nodulation and all further rhizobial evolution was reputed to be explainable only by lateral gene transfer events [17,18].

Analyzing the differences in gene categories and functions in the core genome, and comparing those to a reference full genome, allowed us to test whether the process of core conservation would be due to a simple random gene subsampling (“null” hypothesis) of functional ontology groups, or precise genes would be retained as critical in the core genome. For this comparison, we took as reference the *Rhizobium leguminosarum* bv. *viciae* genome and analyzed its distribution both in the full genome and in the 700 genes core subset shared among the 12 species considered. This comparison rests on the a priori assumption that ontology annotation of orthologous genes should be the same in all of the genome projects analyzed. The core genome resulted in a highly enriched in macromolecule metabolism group, with percentages that increased from 5.64% (full genome) to 20.57% (core genome), and in the metabolism of small molecules that nearly doubles it share. The ribosomal components category was also highly preserved as, out of 55 genes present in the full genome, 43 of those are retained in the core genome, shifting the percentage of this group of genes from 0.75% to 6.14 %.

In the detailed GO terms, as regards cellular components, the membrane item appears strongly depleted in the core. This result applies also to the outer membrane-bound periplasmic space genes. By contrast, the cytoplasm and ribosome domains appear strongly enriched. For the biological processes a strongly lower proportion of conserved genes is shown for the transport and DNA-dependent regulation of transcription complexes, while translation and DNA repair are highly up-represented. The molecular function scheme confirms the under-representation of transport, transcription and DNA binding-related genes, and a corresponding enrichment of those involved in ATP binding, RNA binding and translational machinery.

Overall data suggest a more critical conservation of the downstream aspects of gene expression (protein synthesis) and a looser constraint on the upper sections (mRNA synthesis regulation, inducers, signaling components), whose nature would more flexibly explain the rather diverged ecological context of each taxon.

Coming to the functional divergence analyses, the phenomenon was detected in spite of the orthologous prediction. The observed rate appeared not just a function of available genes, as the percentage of diverged- over available-orthologs varies from a minimum of 1.74% to a maximum of 4.84% depending on the combination of strains examined. Among proteins which resulted under selective functional pressure, we found serine-type endopeptidases, a histidinol-phosphatase activity, glutamate-cysteine ligase activity, all belonging to a context of active metabolic genes. The fact that candidate orthologous attributes by bioinformatic tools could be labelled as functionally diverged in their products functions at the same time is puzzling. In a way, this could fuel the debate known as ‘ortholog conjecture’ [13,14,15,16]. However, in the absence of experimental functional assays, we leave open the possibility that such conflictual definition in a variety of cases could also be due to merely technical issues arising from mismatches in the GO terms definition. This might happen for genomes that have been annotated at different times between which the ontology nomenclature could have undergone updates and refinements.

Summarizing the findings in relation to our hypotheses, the following comments can be made.

The main evidence in terms of involved protein functions were the following: transporters result more prone to variation; our interpretation is that such category is highly involved in different sorts of adaptation and this can bring to host-driven speciation. Support for this consideration can be observed in Figure 4, Figure 5 and Figure 6 in which these functions appear underrepresented in the core genome and are also those that show the highest degrees of divergence.

The Rhizobiales genera considered appear to group in two main clusters based on their genome average identities, one encompassing the *Bradyrhizobium*, *Azorhizobium* and *Mesorhizobium* and the other including *Rhizobium* and *Sinorhizobium*.

In conclusion, the pangenome of the N-fixing symbiotic Rhizobiales proved to be extremely broad in comparison to its core genome. The importance of mobile DNA in the genomics of these bacteria stands out in particular evidence. In this ecologically versatile order, horizontal gene transfer is confirmed to be a major driver of genomic plasticity and a provider of key genes for interactions and finely-tuned adaptations towards plant hosts. The analysis of the core genome allowed us to pinpoint that the majority of its conserved elements belong to an RNA-centered concept, which appears to be critically preserved along the evolution of this functional group of flexibly interactive bacteria.

## Figures and Tables

**Figure 1 microorganisms-09-00407-f001:**
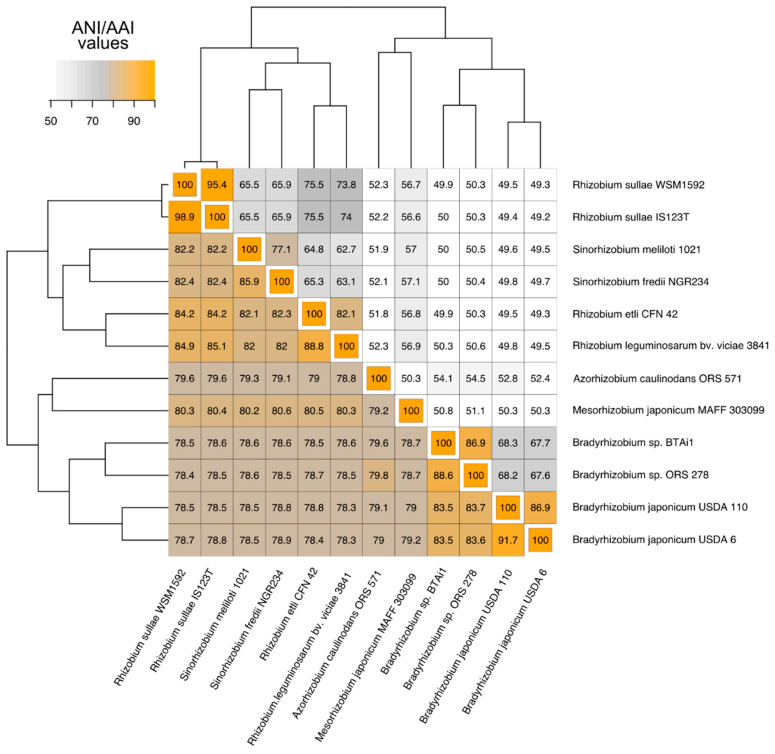
Average nucleotide identity (ANI, lower triangle) and average amino-acid identity (AAI, upper triangle) calculated in pairwise comparisons of the 12 Rhizobia genomes.

**Figure 2 microorganisms-09-00407-f002:**
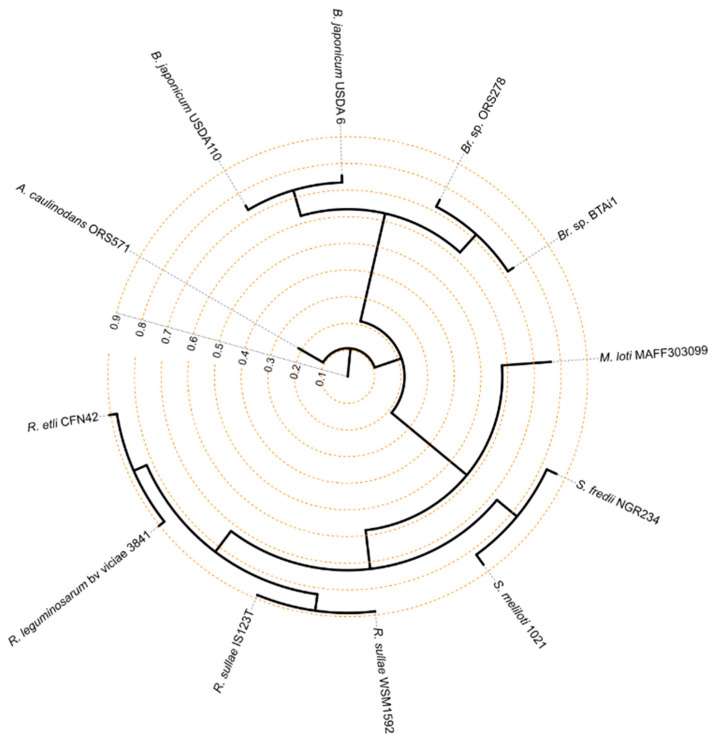
Phylogenomic tree constructed on the basis of the 700 genes forming the core genome of the 12 Rhizobiales tested. *Azorhizobium caulonidans* ORS571, which the ANI indicated as the less similar between Rhizobiales, was used to root the tree. Branch lengths are reported as radial scale departing from the centre of the topology.

**Figure 3 microorganisms-09-00407-f003:**
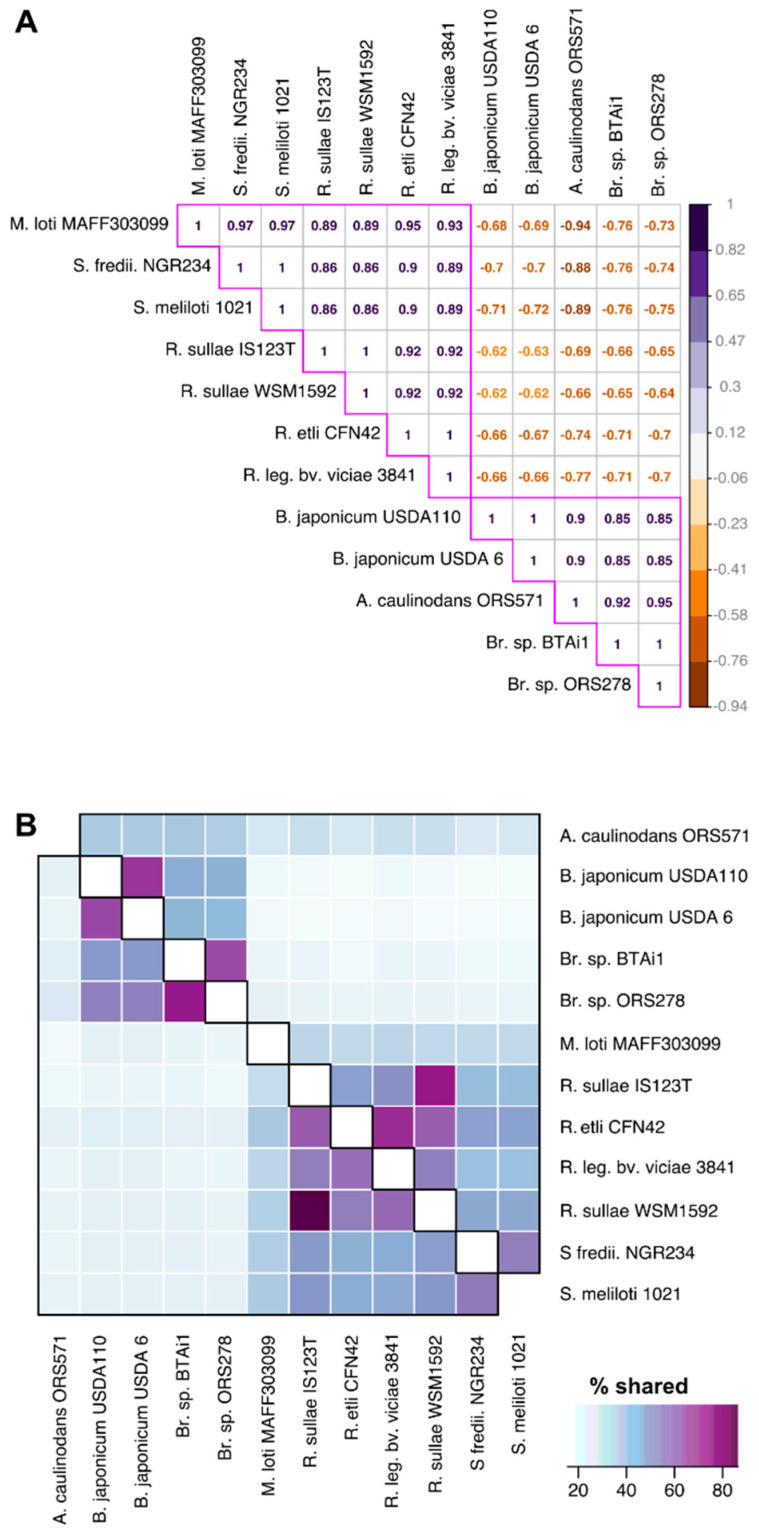
Pairwise (two-taxa) core genome comparisons. (**A**) The heatmap depicts Pearson correlation coefficients computed using the number of shared orthologous genes between pairwise reference genomes. Two main groups can be identified and are here highlighted by pink margins. (**B**) The heatmap shows the genome percentage which is constituted by orthologous genes. The matrix is not symmetric. Reciprocal value differences are contributed by the different number of genes encoded by each reference genome.

**Figure 4 microorganisms-09-00407-f004:**
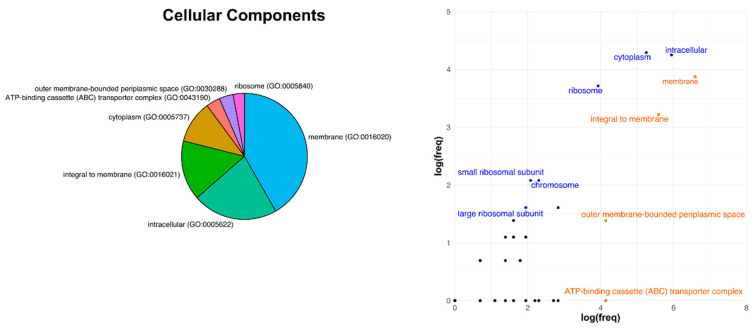
Distribution of Gene Ontology (GO) categories in the full reference genome *R.leguminosarum* bv. *viciae* 3841 and comparisons with the core genome of Rhizobiales as regards the Cellular Components GO terms. Pie-charts on the left-side depict protein functions and relative GO codes belonging to each category and represented at least in the 1% of the whole reference genome. Point charts on the right-side show log transformations of percentage frequencies at the level of whole genome vs. the core genome of Rhizobiales. Positive correlations (>1% difference) indicate enriched categories at the core genome and are shown in blue. Negative correlations (values below –1) indicate depletion of a category from the reference genome, which is higher present in the core genome, and are shown in orange.

**Figure 5 microorganisms-09-00407-f005:**
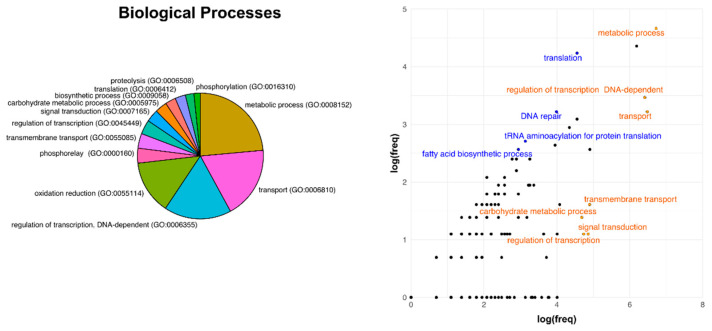
Distribution of Gene Ontology categories in the full reference genome *R.leguminosarum* bv. *viciae* 3841 and comparisons with the core genome of Rhizobiales as regards the Biological Processes GO terms. Descriptive details are the same explained for Figure 4.

**Figure 6 microorganisms-09-00407-f006:**
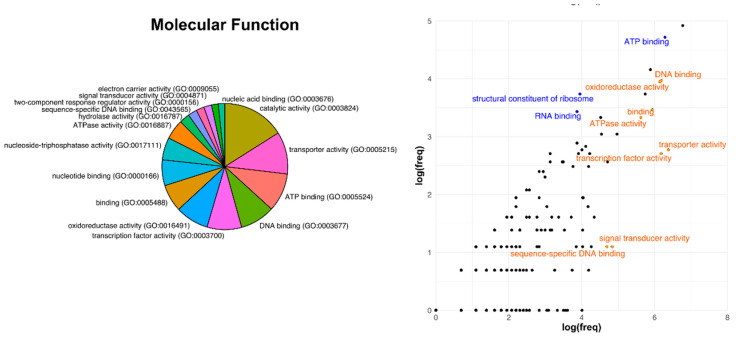
Distribution of Gene Ontology categories in the full reference genome *R.leguminosarum* bv. *viciae* 3841 and comparisons with the core genome of Rhizobiales as regards the Molecular Functions GO terms. Descriptive details are the same as explained in Figure 4.

**Table 1 microorganisms-09-00407-t001:** Comparison of the distribution of genes between chromosomal and plasmid replicons in each individual genome and in the common core genome. N.: Number; nd: not determined.

	Individual Genomes							Core Genome (700)	
	Total Genes	On Chromosome	On Plasmids	N. Plasmids	% On Plasmids	N. Unique Genes	% of Unique Genes in Genome	On Plasmids	% On Plasmids
*A. caul.* ORS571	4780	4780	0	0	0	2239	46.84	0	0
*B. jap.* USDA 110	8374	8374	0	0	0	1908	22.78	0	0
*B. jap.* USDA 6	8888	8888	0	0	0	2424	27.27	0	0
*B. jap.* BtAi1	7694	7466	228	1	2.96	1935	25.15	0	0
*Br.* sp. ORS 278	6783	6783	0	0	0	1354	19.96	0	0
*M. loti* MAFF3030.	7343	6814	529	2	7.20	3804	51.80	0	0
*R. sullae* IS123	7780	nd	nd	nd	nd	1455	18.70	nd	nd
*R. etli* CFN42	6022	4094	1928	6	32.02	1047	17.39	36	5.14
*R. leg*. *viciae* 3841	7342	4797	2545	6	34.66	1796	24.46	37	5.29
*R.sullae* WSM1592	6995	nd	nd	nd	nd	547	8.10	nd	nd
*R. fredii* NGR234	6437	3691	2746	2	42.66	1775	27.57	44	6.29
*S. meliloti* 1021	6287	3425	2862	2	45.52	1686	26.82	50	7.14

**Table 2 microorganisms-09-00407-t002:** Distribution of paralogs in the 12 Rhizobiales genomes. N.: Number

	N. of Paralog Copies									N. of Paralogs	% Paralogs in Genome
	2	3	4	5	6	7	8	11	13		
*A. caulinodans* ORS571	16	0	1	0	0	0	0	0	0	36	0.75
*B. japonicum* USDA110	57	5	1	3	0	0	1	1	1	180	2.15
*B. japonicum* USDA 6	50	4	1	0	0	2	1	0	0	138	1.55
*Br*. sp. BTAi1	55	9	0	2	0	1	0	0	0	154	2.00
*Br*. sp. ORS278	28	3	0	0	0	0	0	0	0	65	0.96
*M. loti* MAFF303099	50	6	1	1	0	1	0	0	0	134	1.82
*R. sullae* IS123T	56	1	0	0	0	0	0	0	0	115	1.48
*R. etli* CFN42	41	2	0	0	3	0	0	0	0	106	1.76
*R. leg.* bv. *viciae* 3841	58	8	2	0	0	0	0	0	0	148	2.02
*R. sullae* WSM1592	29	1	0	0	0	0	0	0	0	61	0.90
*S. fredii* NGR234	65	13	0	5	2	0	2	0	0	207	3.22
*S. meliloti* 1021	49	6	3	0	0	0	0	0	0	128	2.04

**Table 3 microorganisms-09-00407-t003:** Comparison between full genome (*R. leguminosarum* 3841) and the 12-taxa core genome in the gene category distribution.

Category	Whole Genome		Core Genome	
	number	%	number	%
Cell Processes	1478	20.13	100	14.29
Extrachromosomal	130	1.77	0	0
Macromolecule Metabolism	414	5.64	144	20.57
Membrane/Exported/Lipoproteins	664	9.04	28	4.00
Metabolism of Small Molecules	1552	21.14	279	39.86
Regulation	708	9.64	33	4.71
Ribosome Constituents	55	0.75	43	6.14
RNA	61	0.83	0	0
Unknown Function	2280	31.05	73	10.43
Total	7342		700	

**Table 4 microorganisms-09-00407-t004:** Top eight highest-scoring genes in terms of number of different rhizobia in which a significant score of functional divergence was recorded. The handle code shown is that of *B. japonicum* USDA 6.

Gene code (from B.j.USDA6)	Diverged in the Following Rhizobia	Definition	Category	Biological Process	Molecular Function
BJ6T_01910	B. j. USDA110, B. j. USDA 6, B. sp. BTAi1, B.. sp. ORS278, M. l. MAFF303099	Cobyrinic acid synthase	Coenzyme transport and metabolism	undefined	protein binding (GO:0005515)
BJ6T_03330	A. c_ORS571, B. j. USDA110, B. j. USDA 6, B. sp. BTAi1	protease II	Amino acid transport and metabolism	proteolysis (GO:0006508)	serine-type endopeptidase activity (GO:0004252)
BJ6T_04060	B. j. USDA 6, B.sp. BTAi1, B.. sp. ORS278	ATP synthase Gamma chain	Energy production and conversion	ATP synthesis coupled proton transport (GO:0015986)	proton-transporting ATPase, rotational mechanism (GO:0046961)
BJ6T_11840	A.c._ORS571, B. j. USDA110, B. j. USDA 6, B. sp. BTAi1, B. sp. ORS278	DNA polymerase I	Replication, recombination and repair	DNA repair (GO:0006281), DNA replication (GO:0006260)	undefined
BJ6T_16480	B. j. USDA110, B. j. USDA 6, B.. sp. BTAi1, B.. sp. ORS278	inositol monophosphatase family protein	Carbohydrate transport and metabolism	phosphatidylinositol phosphorylation (GO:0046854)	histidinol-phosphatase activity (GO:0004401)
BJ6T_87180	A. c._ORS571 B. j. USDA110, B. j. USDA 6, B. sp. BTAi1, B.. sp. ORS278, M. l. MAFF303099	putative glutathione synthetase	Coenzyme transport and metabolism	glutathione biosynthetic process (GO:0006750)	glutamate-cysteine ligase activity (GO:0004357)
BJ6T_59850	B. j.USDA110, B. j. USDA 6, B. sp. BTAi1, B. sp. ORS278	hypothetical protein	undefined	Cell wall/membrane/envelope biogenesis	undefined
BJ6T_67020	A. c._ORS571 B. j. USDA110, B. j. USDA 6, B. sp. BTAi1, B.. sp. ORS278, M. l. MAFF303099	hypothetical protein	undefined	undefined	undefined

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
