# Peer review of "Pangenomics of the Symbiotic Rhizobiales. Core and Accessory Functions Across a Group Endowed with High Levels of Genomic Plasticity"

_microorganisms, 2021, doi:10.3390/microorganisms9020407_

Round 1

Reviewer 1 Report

This manuscript claim to "challenge the ortholog conjecture paradigm".

The paradigm is shortly introduced, then a number of bioinformatic exercises are carried out and finally it is claimed that "Under these standpoints, the functional divergence analysis of their gene sets provided  additional evidence of the actual variability of genes that are scored as orthologs, contributing to the  further challenging of the Ortholog Conjecture criterion." (l 358-360).

The bioinfomatic exercises carried out are largely based on some auto-pilot analysis, they are poorly documented, and their relevance for the aim is poorly or not explained.

The language is less concise than expected and flawed with non-English sentence structures. An example is "In order to show details of each pairs’ number of shared orthologs. we reported the individual data in the top part of Fig. 3. The dual matrix shown in the bottom part shows instead which percentage of each genome was conserved in the specific set of pairwise shared genes." (l. 208-210)

  • The sentence is passive: "In order to ...
  • a period before "we" should be a comma
  • the numbers in the upper part of Fig. 3 should "show details ..."; they don't show details, they are just numbers of shared genes, the cut-off is not clear; 1E-5 in RBBH or 60 / 60 % ?
  • the bottom part should show a "percentage of the genome" conserved, however it appears that it is not the genome, but just the numbers in the upper part presented in % of all genes.

The questions arising from just these two lines explain why it is too difficult actually to decipher what the authors intend.

Figures are poorly designed and explained; for example in Fig 2 the lengths (should be "distances") are hard to read; boot-straps are just set to "1"; which cannot be correct and this bootstrap is not dealt with further. Such trees should be presented in a way where the length of branches reflects the distances. The impression will then be completely different.

So in summary: the analysis carried out and the data presented appears to have little relevance to the aim of the study and it is hard to find a relevance for the findings in light of the presentation.

Author Response

Reviewer I

This manuscript claim to "challenge the ortholog conjecture paradigm".The paradigm is shortly introduced, then a number of bioinformatic exercises are carried out and finally it is claimed that "Under these standpoints, the functional divergence analysis of their gene sets provided  additional evidence of the actual variability of genes that are scored as orthologs, contributing to the  further challenging of the Ortholog Conjecture criterion." (l 358-360).

ANSWER: In light of this aspect of criticism, the bottom line of the prior version has been completely abandoned and we no longer focus on the ortholog conjecture. We just quote it and we signal the occurrence of genes that qualified as divergent in the comparisons between the genomes that we examined, but we do not claim that such data could strengthen any points concerning that theory. For this reason the previous title (“Pangenomics of the symbiotic Rhizobiales: core genome and functional divergence analyses further challenge the Ortholog Conjecture paradigm”) has been changed and the new one is: “Pangenomics of the symbiotic Rhizobiales. Core and accessory functions across a group endowed with high levels of genomic plasticity”.

The bioinfomatic exercises carried out are largely based on some auto-pilot analysis, they are poorly documented, and their relevance for the aim is poorly or not explained.

ANSWER: besides the change of the main aim, as just outlined, the analyses have been backed up by detailed descriptions, some new ones have been carried out, and all the output files and ensuing graphics have been changed. In practice none of the 7 figures that belonged to the previous version appear in the present one. In addition we also made a new supplementary figures file with three images.  

The language is less concise than expected and flawed with non-English sentence structures. An example is "In order to show details of each pairs’ number of shared orthologs. we reported the individual data in the top part of Fig. 3. The dual matrix shown in the bottom part shows instead which percentage of each genome was conserved in the specific set of pairwise shared genes." (l. 208-210)

  • The sentence is passive: "In order to ...
  • a period before "we" should be a comma
  • the numbers in the upper part of Fig. 3 should "show details ..."; they don't show details, they are just numbers of shared genes, the cut-off is not clear; 1E-5 in RBBH or 60 / 60 % ?
  • the bottom part should show a "percentage of the genome" conserved, however it appears that it is not the genome, but just the numbers in the upper part presented in % of all genes.

ANSWER: the language has been polished and edited throughout the manuscript by a native English speaking person. We have also uploaded a version in which all the parts resulting from changes (amounting to > 50% of the text) have been highlighted in yellow. Incidentally, the portion of text cited by the reviewer is no longer there since a different analysis has been done and the new Fig. 3 has a different comment.

The questions arising from just these two lines explain why it is too difficult actually to decipher what the authors intend. Figures are poorly designed and explained; for example in Fig 2 the lengths (should be "distances") are hard to read; boot-straps are just set to "1"; which cannot be correct and this bootstrap is not dealt with further. Such trees should be presented in a way where the length of branches reflects the distances. The impression will then be completely different.

ANSWER: as anticipated above, all figures of that version have been eliminated from the manuscript and new ones replace them, either as clearer versions stemming from the same analyses but graphically reconstructed as requested or in part as results of new analyses.

So in summary: the analysis carried out and the data presented appears to have little relevance to the aim of the study and it is hard to find a relevance for the findings in light of the presentation.

ANSWER: we have taken this criticism in full consideration and, sticking to the pangenomics evidences, we also  changed the aim of the study which is now centered on the aspect of the wide genomic plasticity that emerges across the pangenome analysis of the symbiotic Rhizobiales, leaving aside the speculative aspects of the possible function divergences in orthologous genes.    

Reviewer 2 Report

Dear authors, I have read with interest your work. The article present important findings on rhizobia for present and future researches. There are some points that need to be addressed in order to improve the article.

Introduction. Please add some more references in order to expand the connections with scientific literature. Also, move the aim of your work in a separate paragraph, and state each objective as a separate sentence. You can add some hypotheses that your work will clarify / elucidate.

Results. When you present abbreviations in tables, add a note bellow table with the entire name of he species. This will make the tables to standalone. 

You need to expand more the presented results. You provide a lot of figures and tables, but their interpretation is too short. There are paragraphs where you speak about more than one figure or table and this makes your article difficult to read. It will be better to make a separate description and interpretation of each figure or table.

Discussion. Please do not cite your figures in the discussion section - this belong to result. You need to discuss your result with reference to other works, and you need to integrate more literature and sources.

Author Response

Reviewer II

 Dear authors, I have read with interest your work. The article present important findings on rhizobia for present and future researches. There are some points that need to be addressed in order to improve the article.

Introduction. Please add some more references in order to expand the connections with scientific literature. Also, move the aim of your work in a separate paragraph, and state each objective as a separate sentence. You can add some hypotheses that your work will clarify / elucidate.

ANSWER: The number of references of this paper has been increased from 39 to 49. The aim has also changed due to the points expressed by Reviewer I, and the new aim and objective is stated at the very end of the Introduction (“The goal of this work was therefore to trace a picture of the balance between gene conservation and gene evolution within a group of bacteria characterized by very dynamic genetics as well as by a broad eco-physiological latitude span.”)

Results. When you present abbreviations in tables, add a note bellow table with the entire name of he species. This will make the tables to standalone. 

ANSWER: We have solved this issue by listing the full name once, and in correspondence with its first mention,i.e. in  the Materials and Methods section.

You need to expand more the presented results. You provide a lot of figures and tables, but their interpretation is too short. There are paragraphs where you speak about more than one figure or table and this makes your article difficult to read. It will be better to make a separate description and interpretation of each figure or table.

ANSWER: this issue has been covered by substituting all prior figures with new ones and implementing their comments in the text. Also a new supplementary material part with three figures has been added.

Discussion. Please do not cite your figures in the discussion section - this belong to result. You need to discuss your result with reference to other works, and you need to integrate more literature and sources.

ANSWER: all figures and tables are now cited only in the results. Five novel references have been introduced in the discussion to comment the observed results.  

Round 2

Reviewer 1 Report

I have still hard to follow the rationale in your work other than for the exercises.

A major point appears to be the "the magnitude of the genetic drift that could have determined a functional variation" (line 365). This analysis is apparently based on current annotations or GO terms or something else, but not on experimental data. as these genomes have been annotated by different ways and at different timepoint it cannot be assumed that an ortholog is annotated as a such.

Another topic is the evidence for horizontal gene transfer, how was this analysed?

I find three versions of section 3.2; should they all be published?

Author Response

Reviewer 1

QUERY: I have still hard to follow the rationale in your work other than for the exercises.

ANSWER: On this point, we feel a possible need for some clarification as the same aspect is apparently driving also the subsequent two issues and many of the ones that had been expressed on the prior version.  This report, as well as the whole special issue of the journal to which it is submitted, (Genomics of Nitrogen-Fixing Plant Symbiotic Bacteria") are devoted to genomics (and not to genetics).

In genomics, and even more in pangenomics, the actual experimental tool is bioinformatics. These studies are done on data (and not on the biological entities from which those data originated).

Therefore, what may seem exercises or auto-pilot analyses or a kind of statistics are instead the work itself.

In this light, we find that the rationale of the matter can be more clearly visualized.     

Having better defined this framework, we can state what was the rationale and which were the main take home messages that emerged from our analyses:

The rationale was to test the degree of genomic divergence in spite of the common, phenotypically unifying, trait of legume nodulation.

The main evidences in terms of involved protein functions were the following: transporters result more prone to variation; our interpretation is that such category is highly involved in different sorts of adaptation and this can bring to host-driven speciation. The support of this consideration can be observed in Figs. 4,5,6 in which these functions appear underrepresented in the core genome and are also those that show the highest degrees of divergence.

The considered Rhizobiales genera appear to group in two main clusters based on their genome average identities, one encompassing the Bradyrhizobium, Azorhizobium and Mesorhizobium and the other including Rhizobium and Sinorhizobium

Parts of these clarifications have been added to the text

QUERY : A major point appears to be the "the magnitude of the genetic drift that could have determined a functional variation" (line 365). This analysis is apparently based on current annotations or GO terms or something else, but not on experimental data. as these genomes have been annotated by different ways and at different timepoint it cannot be assumed that an ortholog is annotated as a such.

ANSWER: As commented above, all this work (and all the other works of this kind) are done in silico (and not in wet lab mode). The actual experimental data are those that the bioinformatics programs do yield upon running them on the datasets. We have added three references to the methods to better drive the attention to this point. In the cited case of functional variation, that was the DIVERGE application, which estimates type-II (cluster-specific) functional divergence of protein sequences. If we were to affirm that computer-based analyses would not be truly experimental, and that their results would not qualify as scientific, no journals or articles of bioinformatics could exist. 

As regards the fact that a number of proteins that score as functionally diverged could be in this sense false positives due to changes in annotation codes nomenclature through time, we definitely agree (we have dropped all the considerations in support of the orthologs conjecture and changed the title of the paper). Moreover we had already stressed this point ourselves in the manuscript’s first revision : (Line 495: “However, in absence of experimental functional assays, we leave open the possibility that such conflictual definition in a variety of cases could also be due to merely technical issues arising from mismatches in the GO terms definition. This might happen for genomes that have been annotated in different times between which the ontology nomenclature could have undergone updates and refinements.”)

If our own sentence (“in absence of experimental functional assays,”) was the point that generated this query, we need to clarify the point. The assays we referred to would belong to studies able to address the loss of function in (available) bacterial strains by generating knock-out mutants through mutagenesis, reverse genetics, phenotypic tests and all the array of in vivo approaches that in the eighties’ age were the standard (and only) way to carry out microbiology studies. One must also keep in mind that those experiments were (and would still be) feasible only one gene at a time in one strain at a time, and requiring year-like durations for the completion of an operon.  But in an -omics setting, consisting in meta-analyses on other authors’ datasets, available in public repositories, those tasks would be out of the boundaries and of the sense of bioinformatics.

If the sensation were that we have not performed enough experimental biology to write this manuscript, we can re-state that for one of the genomes that were analyzed (Rhizobium sullae, IS 123 sp. nov. Squartini 2002) (Ref. 20) we described the novel species, coined its taxonomical name, published several reports on its ecology and physiology in the past 30 years (starting when its name was still Rhizobium hedysari), and finally sequenced its genome (Ref. 31 ) whose genes we are treating in the present report, along with the ones from the other chosen Rhizobiales.

QUERY: Another topic is the evidence for horizontal gene transfer, how was this analysed?

ANSWER: After the above premise we can more clearly explain that the question is to be reversed. The evidence of gene transfer is not a finding from the present report but an established piece of knowledge that stems from a number of independent studies based on classic experimental molecular genetics, (plasmid visualization in agarose gel electrophoresis, Southern blot hybridization, Sanger sequencing of single fragment clones, BLAST alignments etc.) from which the occurrence of insertion elements, larger transposons, traces of illegitimate recombination and gene transfer, were found profusely in rhizobia by many authors (see refs. 7,16,17,27,33,52). We have ourselves contributed to that part of the research on gene transfer evidences for Rhizobium sullae /R. hedysari ( Meneghetti et al. 1996. Plant and Soil 186: 113-120; Muresu et al.  (2005) FEMS Microbiology Ecology 54:445–453.

Then, given these existing acquisitions, in our current analysis we observed how the effects of this known phenomenon appear to be unevenly spread across the different symbiotic Rhizobiales. In particular how Azorhizobium, Mesorhizobium and Sinorhizobium result to have been more homogenized in their core genome by the evolutionary shortcut that lateral gene transfer represents.

QUERY: I find three versions of section 3.2; should they all be published?

ANSWER: The section paragraphs had been mistakenly not given the incremental numeration. We have now inserted the correct sequential series: 3.2; 3.3; 3.4.

Reviewer 2 Report

Dear authors, your work was improved by the additional references and the new form of your sentences. Also, you have expanded the result section which is now in a higher size and better present your findings.

Please, pay attention to the abstract and conclusions - in order to make your findings more visible.

Also, make the aim of your work as a separate paragraph, and state each of your hypotheses that you have demonstrated in the article.

Author Response

Reviewer 2

QUERY: Dear authors, your work was improved by the additional references and the new form of your sentences. Also, you have expanded the result section which is now in a higher size and better present your findings.

ANSWER: We appreciate this acknowledgment.

Please, pay attention to the abstract and conclusions - in order to make your findings more visible.

Also, make the aim of your work as a separate paragraph, and state each of your hypotheses that you have demonstrated in the article.

ANSWER: We have followed the advice and separated the sentences of the aims at the end of the introduction adding text as follows:

“ The goal of this work was therefore to trace a picture of the balance between gene conservation and gene evolution within a group of bacteria characterized by very dynamic genetics as well as by a broad eco-physiological latitude span.”

“In particular, the rationale was to test the degree of genomic divergence in spite of the common, phenotypically unifying, trait of legume nodulation.“

“Our hypotheses were that (a) the genomic plasticity of Rhizobiales would suit the need of tracing an efficient resolution of their differences based on nucleotide and aminoacidic average identities across genomic data and (b) that different functional protein categories would clearly partition as over- or under-represented between the core and the flexible subsets of their genomes, reflecting host specific adaptations.”

While in the discussion we have added this part:

“The main evidences in terms of involved protein functions were the following: transporters result more prone to variation; our interpretation is that such category is highly involved in different sorts of adaptation and this can bring to host-driven speciation. The support of this consideration can be observed in Figs. 4,5,6, in which these functions appear underrepresented in the core genome and are also those that show the highest degrees of divergence.

The considered Rhizobiales genera appear to group in two main clusters based on their genome average identities, one encompassing the Bradyrhizobium, Azorhizobium and Mesorhizobium and the other including Rhizobium and Sinorhizobium. “